# Midwifery care providers' childbirth and immediate newborn care competencies: A cross-sectional study in Benin, Malawi, Tanzania and Uganda

Ann-Beth Moller[1]*, Joanne Welsh[2], Christian Agossou[3], Elizabeth Ayebare[4], Effie Chipeta[5], Jean-Paul Dossou[3], Mechthild M. Gross[2], Gisele Houngbo[3], Hashim Hounkpatin[3], Bianca Kandeya[5], Beatrice Mwilike[6], Max Petzold[1], Claudia Hanson[7,8]

1 School of Public Health and Community Medicine, Institute of Medicine, Sahlgrenska Academy, University of Gothenburg, Gothenburg, Sweden, 2 Midwifery Research and Education Unit, Hannover Medical School, Hannover, Germany, 3 Centre de Recherche en Reproduction Humaine et en Démographie (CERRHUD), Cotonou, Benin, 4 Department of Nursing, College of Health Sciences, Makerere University, Kampala, Uganda, 5 Kamuzu University of Health Sciences, Centre for Reproductive Health, Blantyre, Malawi, 6 Department of Community Health Nursing, School of Nursing, Muhimbili University of Health and Allied Sciences, Dar Es Salaam, Tanzania, 7 Global Public Health, Karolinska Institute, Stockholm, Sweden, 8 Department of Disease Control, London School of Hygiene and Tropical Medicine, London, United Kingdom

* ann-beth.moller.2@gu.se

**Data Availability Statement:** All relevant data are within the paper and its Supporting Information

## Abstract

Evidence-based quality care is essential for reducing sub-Saharan Africa's high burden of maternal and newborn mortality and morbidity. Provision of quality care results from interaction between several components of the health system including competent midwifery care providers and the working environment. We assessed midwifery care providers' ability to provide quality intrapartum and newborn care and selected aspects of the working environment as part of the Action Leveraging Evidence to Reduce perinatal morTality and morbidity (ALERT) project in Benin, Malawi, Tanzania, and Uganda. We used a self-administered questionnaire to assess provider knowledge and their working environment and skills drills simulations to assess skills and behaviours. All midwifery care providers including doctors providing midwifery care in the maternity units were invited to take part in the knowledge assessment and one third of the midwifery care providers who took part in the knowledge assessment were randomly selected and invited to take part in the skills and behaviour simulation assessment. Descriptive statistics of interest were calculated. A total of 302 participants took part in the knowledge assessment and 113 skills drills simulations were conducted. The assessments revealed knowledge gaps in frequency of fetal heart rate monitoring and timing of umbilical cord clamping. Over half of the participants scored poorly on aspects related to routine admission tasks, clinical history-taking and rapid and initial assessment of the newborn, while higher scores were achieved in active management of the third stage of labour. The assessment also identified a lack of involvement of women in clinical decision-making. Inadequate competency level of the midwifery care providers may

files and the data can be made available upon request.

**Funding:** This study is part of the ALERT-project which is funded by the European Commission's Horizon 2020 (No. 847824) under a call for Implementation research for maternal and child health (CH). The contents of this article are solely the responsibility of the authors and do not reflect the views of the EU. Publication fees paid by University of Gothenburg, Sweden (ABM). The funders had no role in study design, data collection and analysis, decision to publish, or preparation of the manuscript.

**Competing interests:** The authors have declared that no competing interests exist.

be due to gaps in pre-service training but possibly related to the structural and operational facility characteristics including continuing professional development. Investment and action on these findings are needed when developing and designing pre-service and in-service training.

**Trial registration:** PACTR202006793783148—June 17th, 2020.

## Introduction

Sub-Saharan Africa (SSA) currently accounts for the highest burden of maternal and newborn mortality and morbidity [1–4] even with an increase in both births attended by skilled health personnel [5] and improved access to facility care during childbirth [6]. The policy assumption that availability and access to childbirth care is sufficient to reduce maternal and newborn mortality may misrepresent any progress made as it does not necessarily reflect the quality of care provided [7]. The majority of maternal and newborn deaths in SSA are preventable and reflect quality of health care provision and inequalities in access to health care as approaches to prevent or manage complications are well known [8–10]. Quality care is necessary to safeguard positive health outcomes [11, 12] and is increasingly recognized internationally as a critical aspect of the unfinished maternal and newborn health agenda [13]. Quality of care is a multi-dimensional and in the field of maternal and newborn care several frameworks have supported shaping actions to improve health outcomes. Among these are the "Quality of care for pregnant women and newborns-the WHO vision" [14] and the "Midwifery and quality care: findings from an evidence-informed framework for maternal and newborn care" developed by Renfrew et al. (2014) focusing specifically on midwifery care [15]. The framework is structured around five components of care: practice categories, organisation of care, values, philosophy, and care providers [15]. Similarly, the framework developed by Mattison et al. (2020) identifies that political and health system components have the ability to both support and hinder the provision of quality childbirth care [16]. The elements of the frameworks are supported by empirical studies. A Cochrane systematic review concluded that the health workforce capacity is often insufficient because of essential deficiencies in pre-service training as well as continuous in-service training [17]. A mapping review of midwifery pre-service curricula in four SSA countries also found gaps in pre-service training when compared to the International Confederation of Midwives (ICM) Essential Competencies Framework. Main gaps identified in all curricula related to women-centred care, inclusion of women in decision making, fundamental human rights of individuals and evidence-based learning. Years of pre-service training, professional titles as well curricula content varied within and among the countries [18]. It is generally acknowledged that globally pre-service midwifery education competency levels are difficult to compare across programmes due to their differing contents and lengths as well as varying faculty capacity including competence of midwifery educators, teaching and learning materials, in addition to the level of clinical training and supervision [19]. Another obstacle to quality care is the lack of an enabling working environment in which to provide care, with even basic commodities such as water, electricity, and transport absent or sporadic in their availability in SSA. Poor supplies of basic medical equipment and general consumables limits the capacity of midwifery care providers to practice to their level of competence [20–22].

Furthermore, a lack of clinical guidelines and poor implementation leave midwifery care providers without standard evidence-based guidance on how to manage both uncomplicated and complex pregnancies [23, 24]. Finally, the high workload, tasks, responsibilities, and the

personal risks to midwifery care providers are often not reflected in their remuneration. All these aspects influence lack of teamwork, collaboration, and trust as well as the interaction with women during childbirth [17].

This study is part of the Action Leveraging Evidence to reduce perinatal mortality and morbidity (ALERT) study. One objective of the ALERT study was to develop an in-service training package for maternity care providers in Benin, Malawi, Tanzania and Uganda. Evidence suggests that in-service training packages available in SSA have a strong focus on emergency obstetric care, with few materials available on antenatal care and standard intrapartum care [25]. The ALERT study aims to address this deficit by creating an in-service training package focusing on basic intrapartum care. This study was designed to inform the content of the ALERT in-service training package [26]. It sought to understand gaps in factors affecting the ability of midwifery care providers to provide quality maternal and newborn care. In particular, it assessed midwifery care providers' competency as well as aspects of the working environment in terms of access of guidelines, equipment and consumables.

## Methods and materials

### Ethical considerations

This study is part of the Action Leveraging Evidence to Reduce perinatal morTality and morbidity in the sub-Saharan Africa project (ALERT). The project received ethical approval from the following institutions:

Karolinska Institutet, Sweden (Etikprövningsmyndigheten—Dnr 2020–01587).

School of Public Health research and ethics committee (HDREC 808) and Uganda National Council for Science and Technology (UNCST)—(HS1324ES).

Muhimbili University of Health And Allied Sciences (MUHAS) Research and Ethics Committee, Tanzania (MUHAS-REC-04-2020-118) and The Aga Khan University Ethical Review Committee, Tanzania (AKU/2019/044/fb).

College of Medicine Research and Ethics Committee (COMREC), Malawi—(COMREC P.04/20/3038).

Comité National d'Ethique pour la Recherche en Santé, Cotonou, Bénin (83/MS/DC/SGM/CNERS/ST).

The Institutional Review Board at the Institute of Tropical Medicine Antwerp and The Ethics Committee at the University Hospital Antwerp, Belgium—(ITG 1375/20. B3002020000116).

The study protocol was published in the Journal of Reproductive Health in 2021 and outlines the methodology for the design and conduct of the assessment of midwifery care providers' childbirth and newborn immediate care competencies [27]. The protocol also outlines a qualitative component pertaining to provider experiences and perception related to in-service training but in this paper, we only report on the results from the quantitative study. Our reporting follows the STROBE Statement for cross-sectional studies [28] (S1 Table).

### Study design

The study used a cross-sectional design employing a self-administered questionnaire for knowledge assessment and skills drills simulation observations to assess skills and behaviours.

The self-administered questionnaire included selected aspects related to the working environment participants were working in i.e., access to guidelines, support, additional training as well as equipment and consumables.

## Study setting

The study was conducted as part of the ALERT study and informed the work stream related to "Positioning Midwifery". The ALERT study aims to evaluate the effect of a four-component intervention on perinatal mortality in 16 hospital maternity units across Benin, Malawi, Tanzania, and Uganda. Selection criteria for these hospitals and characteristics of study countries were previously described in detail [26]. The maternity units were a mix of public and private-not-for-profit, which is common in many SSA countries. Each hospital had a minimum caseload of 2500 births per year. Caesarean section and blood transfusion services were available in all hospitals, and they all have a neonatal care unit except in Malawi. Basic amenities such as electricity and running water were available every day in the maternity units even though electrical power shortages were common. Hand-washing stations and hand cleaning products were sometimes lacking. Although regular sterile gloves were available in all hospitals, long sterile gloves were often missing. Clinical protocols commonly available covered postpartum haemorrhage and (pre-)eclampsia in many of the hospitals. Areas scarcely covered by protocols were sepsis management, postnatal care for the woman and the newborn, standard operating procedure for caesarean section as well as for admissions and transfers out. Most hospitals, except for one in Uganda, had an ambulance service run by the hospitals themselves. However, in most cases they were not meeting reliable standards, relying on the woman and her family to pay for fuel, and serving solely as a means of transport as those driving the ambulances had no formal medical training (data will be openly available through Zenodo (https://www.zenodo.org/) at the end of the project (2024) [29]).

## Target population and sampling

All midwifery care providers including doctors providing midwifery care in the 16 ALERT project hospital maternity units were invited to take part in the knowledge assessment. During the data collection period a strike was on-going in Uganda which affected the number of midwifery care providers present at the hospitals. Approximately one third of the midwifery care providers who took part in the knowledge assessment and working environment assessment were randomly selected and invited to take part in the skills and behaviour simulation assessment.

The definition of a midwifery care provider varies among countries [30]. To ensure clarity, our operational definition of a competent midwifery care provider is described in Box 1.

---

### Box 1. Definition of competent midwifery care providers used in the study

Providers of midwifery care are competent maternal and newborn health professionals educated, trained, and regulated to national and/or international standards. They provide skilled, evidence-based, and compassionate care to women, newborns and families.

Providers of midwifery care:

- Promote and facilitate normal physiological, social and cultural processes throughout the childbearing continuum with a continuity of care philosophy;

- Seek to prevent and manage maternal and newborn complications;

- Consult and refer to other health services where required; and

- Respect women's individual circumstances and views, providing sensitive and dignified care.

Source: [27]

---

## Study tools

The study applied i) a self-administered questionnaire for the knowledge assessment and working environment ii) an observation skills drills checklist for the skills and behaviour simulation assessment. The data collection tools were developed based on the International Confederation of Midwives (ICM) Essential Competencies for Midwifery Practice Framework, 2019 update [31, 32].

The knowledge assessment and working environment questionnaire consisted of seven sections linked to childbirth and immediate newborn care. Section titles are illustrated in Box 2. (the full knowledge assessment questionnaire is included in S1 Text). The tools were assessed for face and content validity by ALERT team members who did not take part in the design of the tools.

---

### Box 2. Content of the sections in the knowledge assessment and the skills drills simulation observation checklist

**Content of the knowledge assessment by sections** (self-administered questionnaire)

Section 1. Provider characteristics (personnel information, educational background and general information related to current job)

Section 2. Working environment (history of in-service training)

Section 3. Triage and referral

Section 4. First stage management of labour

Section 5. Second stage management of labour

Section 6. Third stage management of labour

Section 7. Reporting and documentation, and handover between shifts

The self-administered questionnaire included three questions specifically related to the COVID pandemic.

_______________________________________________________________________________

Content of the skills drill's simulation observation checklist by sections

Section 1. The provider treats the client (Mary) in a cordial manner

Section 2. Admission

Section 3. The provider properly reviews and fills out the clinical history about Mary

Section 4. The provider properly conducts the physical examination between contractions

Section 5. The provider properly conducts the obstetric examination between contractions

Section 6. The provider properly conducts a vaginal examination

Section 7. Documentation

Section 8. The provider assists the woman to have a safe and clean birth

---

Section 9. The provider properly conducts a rapid initial assessment

Section 10. The provider adequately performs active management of the third stage of labour (AMTSL)

Section 11. The provider adequately performs immediate postpartum care

Section 12. Documentation of AMTSL

AMTSL as a prophylactic intervention is composed of a package of three components or steps: 1) administration of a uterotonic, preferably oxytocin, immediately after birth of the baby; 2) controlled cord traction (CCT) to deliver the

placenta; and 3) massage of the uterine fundus after the placenta is delivered.

Ideally, we would have carried out observations of actual clinical care to assess clinical skills and behaviours. However, observations are resource intensive and the assessment procedure could have created tensions and ethical dilemmas in cases where care was being omitted. In view of the co-design nature of the ALERT project we thus opted to carry out the observation study as skills drills simulation sessions which were less threatening.

One ALERT team member who was part of the research team played the "woman" (the woman–called Mary) another the companion (the woman's companion) and were given instructions on how to answer any questions asked by the midwifery care provider using the skills drills simulation script (the skills drills simulation script is included in S2 Text).

An observation checklist was used to record which clinical practices were performed and which were not executed. The checklist included all clinical evidence-based aspects which would be expected to be addressed in each scenario. The skills drills simulation observation checklist contained 12 sections (Box 2). The complete skills drills simulation checklist tool is included in S2 Text. The data collection tools and skills drills simulation scripts were available in English, French and Swahili.

For the skills drills simulations assessment a "Laerdal Mama Birthie Kit" was used [33]. Additional equipment such as a blood pressure machine, thermometer, delivery kit and cloths to use as drapes were available to enhance the simulation and ensuring that all providers had access to the same equipment.

## Pre-test

All the data collection tools were pre-tested in one or two maternity units which were different from the study hospital maternity units in each country but with similar characteristics to the study hospitals i.e., level of the facility, services offered and number of annual deliveries [26]. The pre-tests identified some practical issues related to the conduct of the skills drills simulations such as setting up the room, ensuring all equipment was available, the sequence of sections in the observation checklist as well as some missing aspects in the skills drills simulation script. Tablet electronic devices were tested to ensure accurate programming and for ease of use. With the feedback provided by each country study team the tools were adjusted accordingly.

## Data collection

Data were collected from May to June 2021 in all the 16 hospital maternity units by one ALERT co-investigator with a midwifery background and data collection assistants with

nurse-midwifery training in each country. Standard operating procedures were developed detailing how to carry out the consent procedures as well as how to conduct the knowledge and working environment assessment, and the skills drills simulations together with an equipment list for each activity. The consent form was available in English, French and Swahili. The team member playing the woman (Mary) was the same person for all the four hospital maternity units within the countries. The companion was played by different team members and was not applied in Benin and Tanzania as maternity units currently do not permit women to bring a companion in the labour ward.

Data collection teams underwent training prior to data collection. These sessions responded to the COVID-19 situation at the time and were therefore conducted virtually. Tablet electronic devices were used by participants to fill in the knowledge and working environment assessment questionnaire. For the skills drills simulations one country elected to use the tablet electronic device to fill in the skills drills simulation observation checklist, while the others used a paper format followed by data entry after each skills drill simulation session. The paper format of the checklists was stored in a locked cabinet in the institutions of the project principal investigators. We randomly selected 10 per cent of the paper format checklists and checked against the data entered to the tablets before further processing. We did not find any discrepancies within the data checked. Data from the knowledge and working environment assessment questionnaires and skills drills simulation observation checklist were exported from the tablet electronic devices to the Research Electronic Data Capture (Redcap) (https://www. project-redcap.org/) software and the data were stored on a secure server at the Karolinska Institute, Stockholm, Sweden.

## Data analysis

Datasets were checked, verified, and cleaned with each country team and all data were anonymized prior to analysis. Completed knowledge and working environment assessment questionnaires and skills drills simulation observations data were imported into Stata version 16.0 [34] for analysis. We calculated descriptive statistics, mean or proportion of all variables of interest. No inferential statistics were calculated due to the small study sample size, which would have produced implausibly broad, non-informative, confidence intervals as per the study protocol [27]. No imputation was performed for missing data. In addition to the mean and proportion analysis data from the skills drills simulations were analysed according to performance using a "traffic light" presentation. Red denotes that the task was carried out less than 50% of the time, suggesting a dangerous gap in performance. Yellow denotes that the task was carried out between 50% and less than 80% of the time, suggesting a critical gap in performance. While green suggests that the task was performed in the majority cases to the expected performance level according to recommendations and good practice.

## Results

The study included the 16 ALERT hospital maternity units. From these maternity units 302 out of 414 eligible midwifery care providers took part in the knowledge and working environment assessment assessing routine childbirth and immediate newborn care, and 113 skills drills simulations were conducted to assess skills and behaviour. In Benin 16%, Malawi 40%, Tanzania 16% and Uganda 26% of the midwifery care providers did not take part in the knowledge and working environment assessment as they were not present at the facility during the data collection period due to annual or study leave, not on duty or other personal reasons. In one facility a provider declined to consent and in some cases the providers did not find time to participate (Table 1).

## Participant characteristics

In total 302 participants took part in the knowledge and working environment assessment: Benin 72, Malawi 101, Tanzania 81 and Uganda 48 (Table 1). Table 2 describes participant characteristics in the four countries. Two thirds (74%) of the midwifery care providers were female ranging from 59% in Malawi to 89% in Benin. Overall 64% reported that they have completed tertiary education. About half (51%) were dual trained nurse-midwives, with the majority having obtained a diploma as the highest level of training containing some elements related to the provision of maternity care. For most of the participants years spent in pre-service education were more than two years (92%). Registration with a professional organization was common among all the countries except for Benin where around only half (56%) were registered.

The most common work modality was rotating shifts with very few participants only working day (18%) or night shifts (2%). None of the participants stated working evening shifts only.

## Working environment and knowledge assessment results

Detailed results of the working environment and knowledge assessment by country and for all countries can be found in S2 Table.

Concerning the working environment 30% of the participants replied that they were supervised, while 32% said they were not and 37% reported that they were sometimes supervised. Access to education and training resources varied among the countries with 58% of participants in Benin reporting not having access compared to 15% in Malawi.

Access to equipment and medication needed for caring for women during childbirth differed among the countries. In Benin 49% of participants said equipment and medication were available compared to 73% in Tanzania. 54% of the participants said that their hospital has a written protocol in place for triage related to labour and childbirth, and 45% said that a written protocol is in place for referral during labour and childbirth (only for non-referral facilities). 50% of participants said that no written protocol to assess women with symptoms of COVID-19 was in place and 14% did not know.

Participants were asked about fetal heart rate monitoring during the first stage of labour and how often this should be checked. The majority responded that the fetal heart rate should be checked every 30 minutes and 15% said every 15 minutes (Fig 1), which corresponds to the current recommendations of monitoring fetal heart rate very 15 to 30 minutes during first stage of labour. Regarding fetal heart monitoring in second stage of labour 45% responded that the fetal heart rate should be monitored every 15 minutes and only 14% correctly responded every 5 minutes. [35, 36] (Fig 1).

62% of participants said that they encourage the women to drink and eat as wanted, while 2% said that women should abstain fully from drinking and eating with some country

**Table 1. Total number of midwifery care providers in the study hospital maternity units, number of knowledge assessments and skills drills simulations conducted.**

| Country | Total number of midwifery care providers* | Number/(%) of knowledge assessments | Number/(%) of skills drills simulations |
|---|---|---|---|
| Benin | 86 | 72 (84) | 28 (33) |
| Malawi | 167 | 101 (60) | 39 (23) |
| Tanzania | 96 | 81 (84) | 22 (23) |
| Uganda | 65 | 48 (74) | 24 (37) |
| All countries | 414 | 302 (73) | 113 (27) |

*Number of midwifery care providers at the time of data collection.

**Table 2. Participant characteristics.**

| | All countries | Benin | Malawi | Tanzania | Uganda |
|---|---|---|---|---|---|
| | (n = 302) | (n = 72) | (n = 101) | (n = 81) | (n = 48) |
| **Sex** | | | | | |
| Female | 224 (74%) | 64 (89%) | 60 (59%) | 58 (72%) | 42 (88%) |
| Male | 78 (26%) | 8 (11%) | 41 (41%) | 23 (28%) | 6 (12%) |
| **School education (completed)** | | | | | |
| Primary education | 5 (2%) | 2 (3%) | | 3 (4%) | |
| Secondary education | 105 (35%) | 17 (24%) | 3 (3%) | 58 (72%) | 27 (56%) |
| Tertiary education | 192 (64%) | 53 (74%) | 98 (97%) | 20 (25%) | 21 (44%) |
| **Highest training qualifications related to the provision of maternity care** | | | | | |
| Certificate | 96 (32%) | 47 (65%) | 2 (2%) | 29 (36%) | 18 (32%) |
| Diploma | 139 (46%) | | 78 (77%) | 40 (49%) | 21 (44%) |
| Bachelor | 53 (18%) | 13 (18%) | 21 (21%) | 12 (15%) | 7 (15%) |
| Master/Doctor's degree | 14 (5%) | 12 (17%) | | | 2 (4%) |
| **Year of pre-service training** | | | | | |
| <2 years | 24 (8%) | 1 (1%) | 4 (4%) | 5 (6%) | 14 (29%) |
| >2 years | 278 (92%) | 71 (99%) | 97 (96%) | 76 (94%) | 34 (71%) |
| **Professional title according to pre-service training** | | | | | |
| Nurse-Midwife | 155 (51%) | | 86 (85%) | 69 (85%) | |
| Midwife | 100 (33%) | 60 (83%) | 1 (1%) | 2 (2%) | 37 (77%) |
| Nurse | 1 (0.3%) | 1 (1%) | | | |
| Doctor | 24 (8%) | 7 (10%) | | 10 (12%) | 7 (15%) |
| Other | 22 (7%) | 4 (6%) | 14 (14%) | | 4 (8%) |
| **Years of experience** (n = 300) | | | | | |
| ≤2 yrs. | 86 (29%) | 2 (3%) | 55 (55%) | 22 (27%) | 7 (15%) |
| 3–5 yrs. | 69 (23%) | 5 (7%) | 24 (24%) | 29 (36%) | 11 (23%) |
| 6–10 yrs. | 55 (18%) | 11 (15%) | 14 (14%) | 18 (22%) | 12 (26%) |
| >10 yrs. | 90 (30%) | 54 (75%) | 7 (7%) | 12 (15%) | 17 (36%) |
| **Shifts—the most common** | | | | | |
| Day | 55 (18%) | 27 (38%) | 9 (9%) | 11 (23%) | 8 (10%) |
| Evening | | | | | |
| Night | 7 (2%) | 3 (4%) | 3 (3%) | | 1 (1%) |
| Rotating shifts | 240 (79%) | 42 (58%) | 89 (88%) | 37 (77%) | 72 (89%) |
| **Registered with a professional organization** (n = 299) | | | | | |
| Yes | 244 (82%) | 40 (56%) | 92 (91%) | 66 (81%) | 46 (96%) |
| No | 52 (17%) | 31 (43%) | 4 (4%) | 15 (19%) | 2 (4%) |
| Don't know | 3 (1%) | 1 (1%) | 2 (2%) | | |

variations during first stage of labour. 58% indicated that they strongly encourage companionship throughout labour and birth while 22% pointed out that they do not encourage it as it is not policy in the facility. Almost all (92%) said that they used a partograph if available.

Nearly all participants (95%) said that breastfeeding should begin as soon as possible when the baby is ready within the first hour after birth.

Participants were asked what the APGAR (Appearance, Pulse, Grimace, Activity, and Respiration) letter stands for. 58% provided a correct answer with variation among the countries. Only participants in Benin (38%) indicated that they did not know (Fig 2).

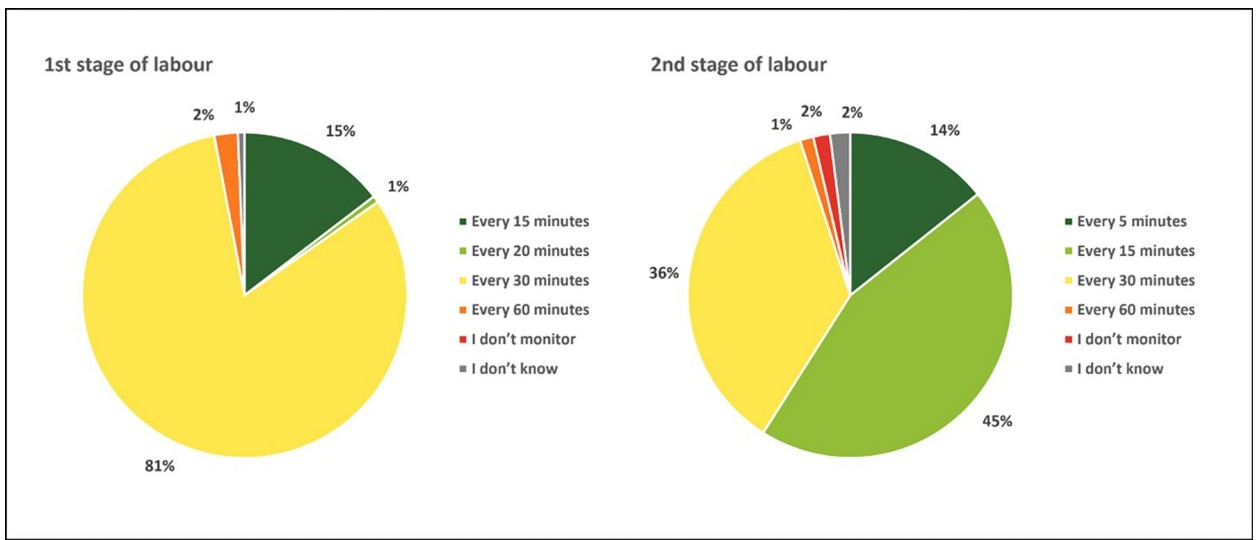

**Fig 1. Respondent's report of how often the fetal heart rate should be monitored during 1st and 2nd stages of labour—all countries.**

Related to monitoring (uterine tone, bleeding, blood pressure and pulse) of women the first two hours after giving birth 63% responded correctly that women should be monitored every 15 minutes, while 32% indicated every 30 minutes.

If the baby is crying and does not need resuscitation, 59% said that the umbilical cord should be clamped or tied within one to three minutes after birth and 35% said immediately after birth (Fig 2).

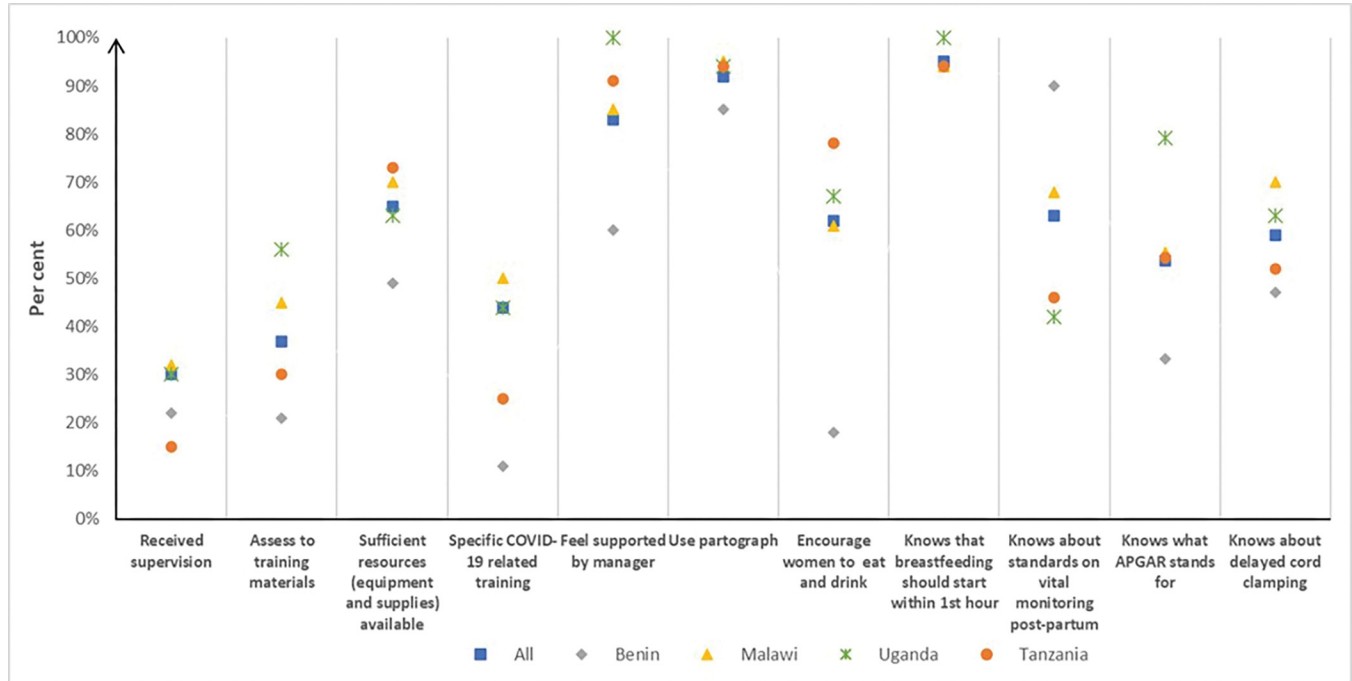

**Fig 2. Key topics covered in the knowledge assessment.**

Almost all participants identified that high blood pressure as well as high levels of protein in the urine were signs and symptoms of postpartum preeclampsia.

Participants were asked about documentation and 4% answered that reporting is done electronically, while the majority used a mix of paper and electronic formats. Furthermore, participants were asked why it is necessary to document and report information and could select multiple answers. The majority of participants (88%) reported that documentation was useful for quality improvement. 69% answered that it is important for colleagues to learn about the women and, that health managers regularly need data (56%), and 28% answered it was done to ensure that the women are informed

The knowledge and working environment assessment questionnaire included three questions related to the COVID-19 pandemic asking about in-service training and availability of personal protective equipment. 56% of the providers reported that they had not received training on how to recognize or assess symptoms of COVID-19 among women with the highest percentage in Tanzania (79%) compared to lowest percentage in Malawi (32%). Only a quarter reported that protective equipment was available to staff in sufficient quantity to change between patients, while 52% reported that equipment was not available in sufficient quantity and 22% reported sometimes.

## Skills drills simulation results

Overall 113 skills drills simulations were conducted: Benin 28, Malawi 39, Tanzania 22 and Uganda 24 (Table 1). Fig 3 depicts the results of the skills drills simulations assessments by sections (1–12). Each section includes the mean performance by country and the overall mean performance for all four participating countries. None of the clinical skills and behaviours assessed reached 100% performance of tasks.

The clinical performance varied by section and across countries. For three of the sections (section 2—admission, section 3—clinical history-taking, and section 9—rapid and initial assessment of the baby), less than 50% of tasks were completed, with small variations across countries. In the other nine sections, the clinical skills and behaviours assessed were all completed by at least half of the participants, with section 7 (documentation [record collected information]) scoring highest with a 71% completion rate. Across all 12 sections of the skills drills simulation, the average score for completing tasks was 61%. Country specific averages for all the 12 sections of the skills drills simulation ranged 42% to 72% (Tanzania 42%, Benin 54%, Malawi 69%, and Uganda 72%) (S1 Fig). Additional skills drills results by country are included in S2 Table.

Further details of each performed task included in the skills drills simulation observation checklist was assessed according to performance level and displayed as a "traffic light" graph (S2 Fig). The result of the assessment suggests that many tasks were performed to the expected level, such as for monitoring fetal heart rate at admission and active stage management of third stage labour. Certain tasks, however, were performed poorly, such as encouraging the woman (Mary) to ask questions, highlighting the lack of involvement of women in decision-making. Assessment of gestational age was lacking (except in Uganda), as was assessment of contractions in addition to monitoring of fetal heart rate in second stage of labour.

## Discussion

The assessment identified strengths as well as areas which need improvement in routine childbirth and immediate postpartum care competencies in all four of the study countries. Almost all participants identified that high blood pressure as well as high levels of protein in the urine were signs and symptoms of preeclampsia. Basic knowledge such as what each letter of

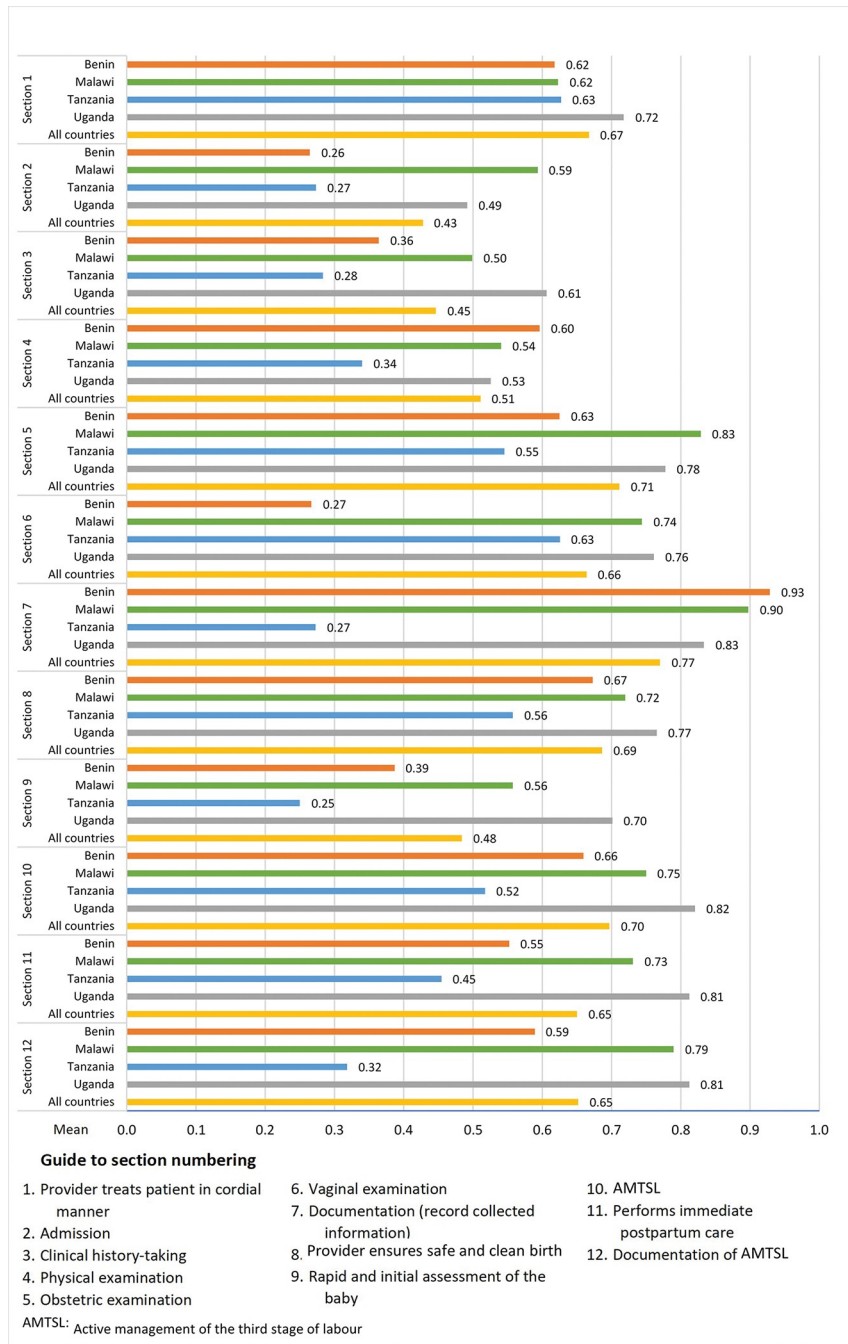

**Fig 3. Skills drills simulations assessment—clinical performance by country and all countries by sections.** Mean of tasks performed within each section.

APGAR stands for were only correctly identified by 58% of the respondents whereas 29% were able to partly provide the correct answer and the remaining respondents did not know. 81% of the respondents correctly reported that fetal heart rate should be monitored at least every 30 minutes in the first stage of labour. Even though Malawi [37] and Tanzania (no reference available) do not have national guidelines on fetal heart rate monitoring during labour, the majority of respondents identified the World Health Organization recommended interval at which

monitoring should be performed. Currently standardized guidelines on fetal heart rate monitoring are lacking as the International Federation of Gynecology and Obstetrics recommends every 15 minutes in the first stage of labour [35] while the World Health Organization recommendation is every 15–30 minutes [36].

Important gaps were identified in relation to admission, clinical history-taking and, rapid and initial assessment of the newborn. The skills drills simulations assessments also pointed to the lack of involvement of women in decision-making. This was illustrated by a failure to encourage the woman (Mary) to ask questions which was only performed by one third of participants (32%). Similarly, clinical findings were only reported back to the woman in 62% of cases. Immediate care of the newborn such as rapid and initial assessment were only performed by 48% of the providers during the skills drills simulations indicating a serious gap.

We identified few studies which assessed knowledge and skills comparable to our study. A study conducted in rural Kenya assessing knowledge of intrapartum care among obstetric care providers, which used a questionnaire, reported similar findings to our study. The mean total knowledge score for all obstetric care providers was 62%, and only 14 providers (4%) scored as 'competent' (a score ≥ 80%) [38]. Underprovided immediate newborn care practices were also acknowledged in a study conducted in health care facilities in Ekiti State, Nigeria essentially due to inadequate knowledge and adherence to evidence-based practices [39].

Another study assessing the quality of midwife-provided intrapartum care in Ethiopia applying multiple data collection methods suggested deficits related to provider competency level, basic infrastructure, lack of equipment and supplies as well as implementation of evidence-based guidelines. Direct comparison cannot be made to our study due to differences in study methodologies but findings from both studies point to the same shortfalls in ensuring competent evidence-based quality of care provision during childbirth and immediate newborn care [40].

Our study participants furthermore indicated a lack of supervision, access to education, training resources and support from their managers. These factors will influence the midwifery care providers' ability to enhance their knowledge and may prevent them from translating knowledge into skills and behaviours, thereby reducing their ability to implement best practice and provide high quality care.

Only one quarter (25%) of participants in our study reported that they had sufficient personal protective equipment to be able to change between patients. This is suggestive that the working environment in which the midwifery care providers work may limit their ability to provide quality care. Findings from our COVID-19 specific questions highlight the importance of the working environment in enabling midwifery care providers to provide quality care. Over half of the midwifery care providers in our study reported they had no training on recognising or assessing for signs of COVID-19, and a quarter of them reported they had no access to sufficient protective equipment to care for women safely.

A study conducted in Kenya, Malawi and India suggests that the quality of intrapartum care women receive is often influenced more by a facility's specific characteristics and the prevailing culture than by the individual midwifery care provider's competencies [41].

These findings may explain why our study countries as well as other SSA countries are struggling to ensure positive health outcomes and meet the Sustainable Development Goals' related to maternal and newborn health [5] as well as the "Strategies toward ending preventable maternal mortality" (EPMM) [42], and the Every Newborn Action Plan (ENAP) [43]. The framework by Mattison et al. (2020) may facilitate the understanding of the political and institutional factors that enable and impede midwifery care providers to perform their tasks [16]. Furthermore, evidence suggests that many low-and-middle income countries are underinvesting in midwifery education despite data suggesting that a reduction in maternal and newborn

morbidity and mortality can be achieved by providing quality midwifery care within an enabling environment [11, 12]. A meta-analysis discussing the theoretical implications for nurse and midwifery educators and decision-makers to enhance the quality of education in Africa suggested several issues. Among the issues identified were outdated pre-service curricula, which do not meet current health system needs; lack of competent teaching staff both for theoretical and clinical teaching as well as inappropriate pedagogical methods and absence of adequate teaching infrastructure [44]. A recent mapping review of pre-service curricula in the ALERT project countries against the International Confederation of Midwives (ICM) Essential Competencies framework [31] suggests that main competence gaps relate to women-centred care, inclusion of women in decision making, fundamental human rights of individuals and evidence-based learning. The review also revealed that in many cases the course literature reading lists were based on outdated textbooks and lacked references to literature on evidence-based guidelines and scientific papers [18].

The findings in our study are not in line with all key health outcomes such as for example the maternal mortality ratio (MMR). Latest maternal mortality estimates suggested that among our study countries Benin had the highest MMR (523 per 100 000 live births in 2020) followed by Malawi (381), Uganda (284) and Tanzania (238) [1]. Tanzania was the country which scored lowest on the skills drills simulation assessment (average of 42% for all the 12 sections) but the country with the lowest estimated MMR for 2020. The MMR estimates are national level estimates and may not reflect the potential differences in maternal mortality at district and facility levels within a country. Based on the SDG target 3.2 "all countries should by 2030 aim to reduce neonatal mortality (NMR) to at least as low as 12 per 1000 live births" [45]. All the study countries are off track to reach this target according to recent estimates for the year 2021. Benin had the highest NMR of 29, followed by Tanzania (20), Malawi and Uganda (19) [2].

Our study findings add to the body of literature reporting midwifery care providers competency gaps. The competency gaps identified in this study will be used to inform the content of the in-service training packages as part of the ALERT study.

We advocate that other stakeholders address these gaps when designing in-service training programmes for midwifery care providers.

## Strength and limitations

The strengths of this study are that we were able to assess not only knowledge but also skills and behaviours using simulation of midwifery care providers across the four project countries. The limitations include the inability to assess all aspects of childbirth and immediate newborn care according to the ICM Essential Competencies, not being able to carry out direct observation to assess skills and behaviours, as well as the relatively small sample of participants. The knowledge and working environment assessment only covered selected topics related to routine childbirth and immediate newborn care. The study tools underwent face and content validation but further validation such as internal consistency and test-retest reliability were not performed. The skills drills simulations had several limitations. The skills drills simulations were conducted by different country teams which may not have performed the sessions in exactly the same way even though a standard operating procedure detailing how to perform the sessions was available. All needed equipment was provided as it was not possible to conduct the skills drills simulations in the maternity units but the set-up provided the opportunity to assess the individual providers competency across the countries. The team members observing and completing the skills drills simulations observation list may have had differing views on the level to which the specific tasks were performed by participants leading to potential

inter-observer disagreements between countries. Given that the study was carried out in diverse settings, the social determinants of health, cultural aspects related to childbirth, and the local health workforce may have influenced responses to the assessments. Due to the diversity of settings, it may not be possible to generalize the study's findings but likely that similar deficiencies would also be observed in midwifery care providers in other similar settings.

## Conclusion

Our study findings revealed gaps in competencies among midwifery care providers as well as gaps in the working environment in all the study hospitals in Benin, Malawi, Tanzania and Uganda. The important gaps identified warrant additional research with larger sample sizes and in more diverse settings to confirm or disprove our findings. Further research will also be crucial to investigate if the gaps relate to inadequate pre-service training, lack of continued professional development, specific facility characteristics, and/or the cultural and social environment. However, our study findings indicate that improvements are warranted due to midwifery care providers competency gaps related to routine childbirth and immediate newborn care, lack of essential commodities, evidence-based guidelines and mentorship. To overcome these limitations and identify pertinent solutions more research is required and substantial investments in midwifery care providers education and training, health care facility infrastructure and quality improvement strategies are needed to ensure that all women, newborns and families are provided with quality midwifery care during childbirth and the immediate newborn period.

## Supporting information

**S1 Fig. Skills drills simulations summary results.**
(DOCX)

**S2 Fig. Skills drills simulations traffic light.**
(DOCX)

**S1 Table. STROBE statement.**
(DOCX)

**S2 Table. Knowledge and working environment assessment results section 2–7.**
(DOCX)

**S1 Text. Knowledge assessment and working environment questionnaire.**
(DOCX)

**S2 Text. Skills drills simulation script and observation checklist.**
(DOCX)

## Acknowledgments

We would like to thank all the midwifery care providers in the ALERT project maternity units who took part in the knowledge assessment and the skill drills simulation sessions.

## Author Contributions

**Conceptualization:** Ann-Beth Moller.

**Data curation:** Elizabeth Ayebare, Effie Chipeta, Gisele Houngbo, Hashim Hounkpatin, Bianca Kandeya, Beatrice Mwilike.

**Formal analysis:** Ann-Beth Moller.

**Funding acquisition:** Max Petzold, Claudia Hanson.

**Methodology:** Ann-Beth Moller.

**Project administration:** Claudia Hanson.

**Supervision:** Max Petzold, Claudia Hanson.

**Writing – original draft:** Ann-Beth Moller.

**Writing – review & editing:** Joanne Welsh, Christian Agossou, Elizabeth Ayebare, Effie Chipeta, Jean-Paul Dossou, Mechthild M. Gross, Gisele Houngbo, Hashim Hounkpatin, Bianca Kandeya, Beatrice Mwilike, Max Petzold, Claudia Hanson.

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
