## [Decision Letter · Decision Letter 0]

4 Jan 2023

PGPH-D-22-01875

Midwifery care providers’ childbirth and immediate newborn care competencies: a cross-sectional study in Benin, Malawi, Tanzania and Uganda

Dear Dr. Moller,

Thank you for submitting your manuscript to PLOS Global Public Health. After careful consideration, we feel that it has merit but does not fully meet PLOS Global Public Health’s publication criteria as it currently stands. Therefore, we invite you to submit a revised version of the manuscript that addresses the points raised during the review process.

We look forward to receiving your revised manuscript.

Kind regards,

Kaveri Mayra

Academic Editor

Journal Requirements:

2. Some material included in your submission may be copyrighted. According to PLOS’s copyright policy, authors who use figures or other material (e.g., graphics, clipart, maps) from another author or copyright holder must demonstrate or obtain permission to publish this material under the Creative Commons Attribution 4.0 International (CC BY 4.0) License used by PLOS journals. Please closely review the details of PLOS’s copyright requirements here: PLOS Licenses and Copyright. If you need to request permissions from a copyright holder, you may use PLOS's Copyright Content Permission form.

Potential Copyright Issues:

Fig 1: Please confirm (a) that you are the photographer; or (b) provide written permission from the photographer to publish the photo(s) under our CC-BY 4.0 license.

Reviewers' comments:

Reviewer's Responses to Questions

**Comments to the Author**

1. Does this manuscript meet PLOS Global Public Health’s publication criteria? Is the manuscript technically sound, and do the data support the conclusions? The manuscript must describe methodologically and ethically rigorous research with conclusions that are appropriately drawn based on the data presented.

Reviewer #1: Yes

Reviewer #2: Yes

2. Has the statistical analysis been performed appropriately and rigorously?

Reviewer #1: Yes

Reviewer #2: Yes

3. Have the authors made all data underlying the findings in their manuscript fully available (please refer to the Data Availability Statement at the start of the manuscript PDF file)?

Reviewer #1: Yes

Reviewer #2: Yes

4. Is the manuscript presented in an intelligible fashion and written in standard English?

Reviewer #1: Yes

Reviewer #2: Yes

5. Review Comments to the Author

Reviewer #1: Thank you for giving me the opportunity to review this paper. This is a solid study; congratulations to the authors.

I think it needs better framing to situate it within the literature, particularly on methods. I am not suggesting it needs additional analysis.

I also think the conceptual framing needs to be more precise to make the implications clearer. You measure knowledge and skills/behaviours (which together make up competency); you also measure perceptions of some things that will influence competencies (e.g. access to education and training resources); you also measure some other things that will influence how competencies will translate into actual practice (e.g. PPE). Likewise in the intro and discussion you talk about how the evidence base shows that pre-service education (PSE) tends to be weak; that competencies tend to be weak; how quality of care tends to be weak; and how there are many other things that influence quality of care aside from provider competency. But nowhere do you give a tight, clear explanation of how all these things fit together. Often you bounce around between these things within paragraphs. As a result, there isn’t a clear narrative running through the study. If you got your up front framing clear and precise, then you would be able to maintain this throughout the manuscript, which would improve it immensely.

One of the things that I wasn’t entirely clear on – which will influence several comments below – is whether “success” in the skills drills was entirely down to the provider. Page 3 of S3 makes it seem like it is ensured that all of the equipment that they need is ready before they do the skills drills; in which case, any deficiencies must solely be due to the actions of the provider. However, if this is not the case, and some of the sub-optimal skills practices could be caused by facility level deficiencies (e.g., not having the requisite infrastructure and supplies), this has very different interpretations for your study. I’m assuming the former, that the skills drill is a measure of individual competency.

Abstract

- Line 30: I think you need to make very explicit that when you use the word ‘behaviours’, you mean it in the context of the skill drill simulations, and not in term of actual practice. I think you also need to make explicit in the abstract that a skills drill relates to a simulation.

- Line 39-42. In line with my overall comment, I think you need to be clearer on what factors are other contributors to knowledge and skills aside from pre-service education; and what are the factors that may constrain competencies from translating into practice. You don’t delineate those sufficiently. You should come back and edit these lines after addressing them in the main text.

Background

- Line 73 typo – to what extent

- Line 68 onwards. You are (I think) essentially arguing that: 1) the competency (knowledge+skills) of the health workforce is not good enough to provide quality care (and that this is partially due to poor pre-service education), and 2) there are many other factors that affect how healthworker competency translates into quality care. But you jump between these two things without making them explicit. The flow of this para is hard to follow as a result (e.g., see the sentence that starts on line 83 – seems in a strange place). I think you need to set out a clearer framework of how everything fits together. You mention the Renfrew framework – this may be a good one (although you also mention but don’t explain Mattison on line 418). If you get this framing stronger, the narrative flow through your results and discussion will be much sharper.

- I think you need to be much more explicit on what the novel aspect of your study is, and how it contributes to the evidence base. This will be easier when your framing earlier was clearer. You make an argument that the current empirical evidence on routine care competency (as opposed to EmONC) is sparse. I would have liked to see more justification of this. I’m really surprised by the assertion that there are not more studies on the competencies of providers of routine delivery and newborn care in these countries.

- I have suggested later bringing in lines 423-434 from your discussion (on how current PSE is weak) here, as a motivator for your study. If you want to argue that you are doing the study to contribute to better PSE which will contribute to better competency which will contribute to better QoC (although there are many other factors affecting it), then this would be a coherent justification.

- I think you should set out more context from each of the four countries of study, and the evidence base (briefly) on QoC and competencies from all of them. You should also come back to this in the discussion.

Study Setting

- I found lines 132 onwards potentially misplaced. Where is this data coming from? Should this not be in the results? Why have you chosen these pieces of information? How do they relate to average places of delivery in these countries? This section overall needs more thought.

Study Tools

- It would be nice to reflect on these tools; how similar tools have been used in other settings/are they novel etc. What are the strengths and limitations of using these kinds of tools? See a recent study in India that used similar approaches - Where there is no nurse: an observational study of large-scale mentoring of auxiliary nurses to improve quality of care during childbirth at primary health centres in India - PMC (nih.gov) – and the references cited within (40 onwards). You should try to situate your study within the methodological literature.

Pre-Test

- There are various typos here: proof read carefully.

Knowledge assessment results p. 15

- Lines 265- are about background characteristics; I would make this a separate section.

- Line 283 onwards – this is where I think your lack of precision in the intro doesn’t help the clarity here – you mention lots of different things that impact different parts of your study/implicit framework, so what you are actually showing here isn’t clear. You should split this into more discrete sections, and refer back to an overall framework. Many of these facts are included in Figure 3 which isn’t referenced here. Some of these overlap with the Study setting section.

o Line 283 is about equipment availability: about the ability to translate competency into practice (which you do not measure).

o Line 285 is about written protocols. What is the logical implication of this? Do we assume? that not having a written protocol will undermine skills/behaviour? Or knowledge? Or both?

o Line 289 is actually about knowledge. But what is the right answer? Is there an objective right answer? Later on you talk about different countries having different policies – how does this affect here?

o Line 302. Is there a right answer? Or does it depend on country policies. How do we interpret this?

o Some of these (e.g. use of partograph line 304) are about clinical practice not knowledge. Even accompaniment is about what actually goes on, not necessarily knowledge. (I hope this gives a concrete example of why you need a clearer framework and more precision). Likewise line 318 on reporting – this is about actual practice not knowledge.

o Lines 325 onwards are definitely not about knowledge!

Skills drills results

- Line 346-349 on overall averages – good to include these in Figure 4, not just the supplemental file.

Discussion

- I think you should try to make the connection between knowledge and skills – to what extent are some of the weak simulated skills and behaviours due to knowledge defects? Can you draw threads through the two sets of data?

- Overall, this would all be much sharper if you had more precision up front. For example, Line 377: presumably supervision, education and training resources, and managerial support are things that impact retention of knowledge and skills/new learning – so are reasons why competencies are partial; it’s not just about poor PSE. Yet Line 378 you start talking about PPE, which won’t affect any of the things you have measured – and will be one of the things that affect the transmission between competencies and practice (which you don’t measure), which you need to reflect on.

- Line 380 – what does the fact that there are different guidelines mean for the interpretation of this study?

- Line 390 – the evidence base should also have been referred to in the introduction (see my comment there). Some of them (e.g. lines 401-405), it’s not clear whether they are measuring competencies or actual practice; the text needs more precision.

- Again you mix literature findings on i) low competencies, ii) weak PSE, iii) other factors that constrain the translation of competencies into practice – without weaving these into a narrative story about what your findings tell us.

- The Mattison framework is not explained (line 418). Again you bounce between this point (factors undermining providers) to the next sentence (line 420) about lack of investment in education. The narrative flow needs to be stronger.

- Line 423-434 – this point about low quality of education – you should bring this into the introduction as a motivator of your study. You do not assess the quality of education in your study, so it does not fit as well in your discussion (but would be important to bring into your introduction).

- Come back to the country level variations; some comment here. Are these in line with outcomes in these countries (representativeness of your sample issues aside).

Strengths and limitations

- You should reflect more on the limitations of only measuring simulated skills and behaviours, and not also actual practice, and the evidence base on these types of methods (See earlier comment).

Reviewer #2: Peer review comments: Midwifery care providers’ childbirth and immediate newborn care competencies: a cross-sectional study in Benin, Malawi, Tanzania, and Uganda

General comments

Thank you for undertaking this complex research work. The result of the study is a good starting point to evaluate the competencies of midwives regarding labor and childbirth. Although this study is already informative in itself, the study will motivate more research work around the evaluation of the competencies of midwives in providing labor and childbirth beyond EMONC

Title

The title is passive.

The study contexts (population and sites) are far away from each other. Consider rewording

Abstract

How did you sample the participants?

What analyses did you perform?

Background:

The study rationale, significance, and purpose are clearly stated.

Are there any previous studies documenting the competencies of care providers with respect to childbirth and immediate newborn care?

Methods

How did you recruit the study participants?

What sampling techniques did you use to sample your participants?

Did you test for the validity of the tools that you developed? If so, how did they perform?

How long did it take to administer the questionnaires?

Results

The results are robust and highly informative. Thank you

Discussion

Most of the glaring study limitations are well highlighted, thank you.

If your findings are not generalizable, can they be transferrable given that your study is purely quantitative?

6. PLOS authors have the option to publish the peer review history of their article (what does this mean?). If published, this will include your full peer review and any attached files.

**Do you want your identity to be public for this peer review?** For information about this choice, including consent withdrawal, please see our Privacy Policy.

Reviewer #1: **Yes: **Thomas Newton-Lewis

Reviewer #2: No

---

## [Decision Letter · Decision Letter 1]

6 Mar 2023

PGPH-D-22-01875R1

Midwifery care providers’ childbirth and immediate newborn care competencies: a cross-sectional study in Benin, Malawi, Tanzania and Uganda

Dear Dr. Moller,

Thank you for submitting your manuscript to PLOS Global Public Health. After careful consideration, we feel that it has merit but does not fully meet PLOS Global Public Health’s publication criteria as it currently stands. Therefore, we invite you to submit a revised version of the manuscript that addresses the points raised during the review process.

We look forward to receiving your revised manuscript.

Kind regards,

Kaveri Mayra

Academic Editor

Journal Requirements:

Additional Editor Comments (if provided):

Reviewers' comments:

Reviewer's Responses to Questions

**Comments to the Author**

1. If the authors have adequately addressed your comments raised in a previous round of review and you feel that this manuscript is now acceptable for publication, you may indicate that here to bypass the “Comments to the Author” section, enter your conflict of interest statement in the “Confidential to Editor” section, and submit your "Accept" recommendation.

Reviewer #1: (No Response)

Reviewer #2: All comments have been addressed

2. Does this manuscript meet PLOS Global Public Health’s publication criteria? Is the manuscript technically sound, and do the data support the conclusions? The manuscript must describe methodologically and ethically rigorous research with conclusions that are appropriately drawn based on the data presented.

Reviewer #1: Yes

Reviewer #2: Yes

3. Has the statistical analysis been performed appropriately and rigorously?

Reviewer #1: Yes

Reviewer #2: Yes

4. Have the authors made all data underlying the findings in their manuscript fully available (please refer to the Data Availability Statement at the start of the manuscript PDF file)?

Reviewer #1: Yes

Reviewer #2: Yes

5. Is the manuscript presented in an intelligible fashion and written in standard English?

Reviewer #1: Yes

Reviewer #2: Yes

6. Review Comments to the Author

Reviewer #1: This article is looking very strong, and the previous comments have been incorporated well. Many congratulations to the reviewers.

A few minor comments:

- A couple of full stops are missing after the edits (lines 33, 71, 208, 435). 465 has one that is not needed. Just do a final proofing.

- Lines 298-308 – which are about working environment – should come in the section starting line 311 (titled knowledge and working environment assessment results) rather than the participant characteristic sections (particularly as the data is reported in figure 1 and not table 2).

- Line 315 (and around): I still feel you could help a non-medical reader like myself by saying “the majority responded correctly that…”. Not saying what is correct or not correct makes it hard for the reader to interpret the information. In the discussion section, you are explicit about this (you use ‘correct’ on line 399) but it would be good to also have it here. If there are areas where what is correct is not obvious but the issue is inconsistency, then I would make that explicit (in line with your response to my earlier comment).

- Line 347-348 – this writing is confusing, just have another look at the sentence

- Is line 427 (participants further…) about your study or about the Ethiopia study introduced on line 421? Because it is not a new paragraph, it is not clear. I assuming it is about your study and the start of a new point – in which case, bring it to a new para and signpost it accordingly. I.e. make it clear that your study also identified some of the working environment challenges that midwives faced which will compromise QoC. I still think you could be more precise that these are both factors that will prevent midwives from increasing their knowledge (access to training resources, supervision) and factors that will prevent them from translating their knowledge into QOC (e.g., lack of equipment and supplies).

- The COVID-19 text (431-435) – it feels a bit strange to introduce it in the discussion, because you did not describe it in the findings (apart from some of the tables). I think you should either mention it also in the findings section and keep it here, or remove it from here.

- 474 limitations not limitation. In this section, did you want to reference back to the challenge of not including DODs to assess practice?

- Line 506 and 507 (we should do this) seem to clash with 501-502 (we should do more research to work out what to do) – just think about rewording this slightly?

Reviewer #2: I thank the authors for addressing the previous comments.

However, a second look into the methods indicate that the study group is so diverse (doctors, midwives, nurses, nurse-midwife). The WHO/FIGO/ICM came up with the terminology 'skilled health personnel' to describe this diverse groups.

I suggest the authors adopt this terminolgy for this paper.

7. PLOS authors have the option to publish the peer review history of their article (what does this mean?). If published, this will include your full peer review and any attached files.

**Do you want your identity to be public for this peer review?** For information about this choice, including consent withdrawal, please see our Privacy Policy.

Reviewer #1: **Yes: **Thomas Newton-Lewis

Reviewer #2: No

---

## [Decision Letter · Decision Letter 2]

17 Apr 2023

Midwifery care providers’ childbirth and immediate newborn care competencies: a cross-sectional study in Benin, Malawi, Tanzania and Uganda

PGPH-D-22-01875R2

Dear Dr. Moller,

We are pleased to inform you that your manuscript 'Midwifery care providers’ childbirth and immediate newborn care competencies: a cross-sectional study in Benin, Malawi, Tanzania and Uganda' has been provisionally accepted for publication in PLOS Global Public Health.

Best regards,

Dr. Kaveri Mayra

Academic Editor

Reviewer Comments (if any, and for reference):

Reviewer's Responses to Questions

**Comments to the Author**

1. If the authors have adequately addressed your comments raised in a previous round of review and you feel that this manuscript is now acceptable for publication, you may indicate that here to bypass the “Comments to the Author” section, enter your conflict of interest statement in the “Confidential to Editor” section, and submit your "Accept" recommendation.

Reviewer #1: All comments have been addressed

Reviewer #2: All comments have been addressed

2. Does this manuscript meet PLOS Global Public Health’s publication criteria? Is the manuscript technically sound, and do the data support the conclusions? The manuscript must describe methodologically and ethically rigorous research with conclusions that are appropriately drawn based on the data presented.

Reviewer #1: Yes

Reviewer #2: Yes

3. Has the statistical analysis been performed appropriately and rigorously?

Reviewer #1: Yes

Reviewer #2: Yes

4. Have the authors made all data underlying the findings in their manuscript fully available (please refer to the Data Availability Statement at the start of the manuscript PDF file)?

Reviewer #1: Yes

Reviewer #2: Yes

5. Is the manuscript presented in an intelligible fashion and written in standard English?

Reviewer #1: Yes

Reviewer #2: Yes

6. Review Comments to the Author

Reviewer #1: You have incorporated the previous comments well, many congratulations. A couple of very very minor comments:

- Line 67 - I think this is missing a word (multi-dimensional xxx)

- Line 57-91 is one very long para, consider breaking it into several

- Line 112-113 - you already introduced the acronym lines 98-99

- Line 316 every not very

- Line 340 - should this be Figure 2 not Figure 3? I'm not sure.

Reviewer #2: None

7. PLOS authors have the option to publish the peer review history of their article (what does this mean?). If published, this will include your full peer review and any attached files.

**Do you want your identity to be public for this peer review?** For information about this choice, including consent withdrawal, please see our Privacy Policy.

Reviewer #1: **Yes: **Thomas Newton-Lewis

Reviewer #2: No
